# Association of Internet Use with Attitudes Toward Food Safety in China: A Cross-Sectional Study

**DOI:** 10.3390/ijerph16214162

**Published:** 2019-10-28

**Authors:** Jiaping Zhang, Zhiyong Cai, Mingwang Cheng, Huirong Zhang, Heng Zhang, Zhongkun Zhu

**Affiliations:** 1School of Economics and Management, Tongji University, Shanghai 200092, China; zhangjp0626@tongji.edu.cn (J.Z.); walkerzhang@tongji.edu.cn (H.Z.); 2School of Management and Labor Relations, Rutgers University, New Brunswick, NJ 08854, USA; zh239@scarletmail.rutgers.edu; 3Commission of Student Affairs, Nanjing Audit University, Nanjing 211815, China; 213421@nau.edu.cn; 4School of Labor Relationship, Shandong Management University, Jinan 250357, China; 5College of Economics & Management, Huazhong Agricultural University, Wuhan 430070, China; zzk@mail.hzau.edu.cn

**Keywords:** food safety, Internet use, food safety perception, food safety evaluation, China

## Abstract

A growing body of research has shown that people’s attitudes toward food safety is affected by their availability and accessibility to food risk information. In the digital era, the Internet has become the most important channel for information acquisition. However, empirical evidence related to the impact of Internet use on people’s attitudes towards food safety is inadequate. In this study, by employing the Chinese Social Survey for 2013 and 2015, we have investigated the current situation of food safety perceptions and evaluations among Chinese residents and the association between Internet use and individuals’ food safety evaluations. Empirical results indicate that there is a significant negative correlation between Internet use and people’s food safety evaluation in China. Furthermore, heterogeneity analysis shows that Internet use has a stronger negative correlation with food safety evaluation for those lacking rational judgment regarding Internet information. Specifically, the negative correlation between Internet use and food safety evaluations is more obvious among rural residents, young people, and less educated residents. Finally, propensity score matching (PSM) is applied to conduct a robustness check. This paper provides new evidence for studies on the relationship between Internet use and an individuals’ food safety cognition, as well as additional policy enlightenment for food safety risk management in the digital age.

## 1. Introduction

Improving food safety not only has a direct contribution to consumers’ health, but is also a vital guarantee toward promoting social stability and sustainable economic development [1,2,3,4,5,6,7,8]. Over the past few decades, despite the stricter and more standardized supervision of food safety around the world [9], public discussions and concerns about food safety have been persisting [10]. In particular, because consumers are the direct victim of foodborne diseases, their perception and evaluation of food safety have attracted increasing attention from scholars [11,12,13,14,15,16,17,18].

A growing body of evidence suggests that people’s food safety perceptions are affected by their availability and accessibility to food risk information [19,20,21,22]. Therefore, media coverage is regarded as one of the most effective channels to acquire risk information and has become an important factor affecting people’s food safety perceptions and evaluations [23,24,25], and has also become a major data source to conduct food-safety-related studies in recent years (e.g., Park et al. [26] and Holtkamp et al. [27]). Scholars in this field have maintained a sustained interest in investigating whether media coverage affects public perception and evaluation of food safety risks. Most relevant studies have found that media scandal reports can enhance people’s awareness of food safety risks in general. For example, Liu and Ma [25] found that people with media exposure are more concerned about food safety risks via a quantitative analysis of data from citizen surveys in 30 Chinese cities.

Nowadays, with the rapid development and application of Internet technology, human beings have entered the digital society. According to the international telecommunication union (ITU), by the end of 2018, the number of Internet users worldwide had reached 3.9 billion [28]. The Internet has become one of the most preferred channels for people to obtain information, greatly promoting the interconnection of the world, making it possible to communicate remotely, and enhancing the scope and intensity of social interactions [29,30,31,32]. Therefore, the online world is bound to become an important place for governments, social institutions, and organizations to disseminate food safety knowledge and strengthen the food safety awareness of the public. Moreover, existing research has also proved that the Internet has indeed had some positive effects on food safety management, e.g., Zhu et al. [23] and Peng et al. [33]. However, unlike traditional media (e.g., newspapers and television), which are strictly controlled and censored by the authorities, the Internet has facilitated the “privatization” and “fragmentation” of mass media. In other words, the Internet is more flexible in reporting food safety related information (negative and positive). As a result, the Internet may also promote the spread of food safety rumors and amplify public concerns about food safety [25].

In addition, “negativity bias” is a common phenomenon in psychology and the social transmission of information, where people instinctively have a stronger preference for negative news or events. Compared with positive and neutral stimuli, negative stimuli have a more lasting impact on people’s emotions [34,35,36,37,38]. Therefore, in order to get more clicks, web media or web editors may be more inclined to report food safety related scandals and negative news. Moreover, due to netizens’ innate interest in negative news, negative news related to food safety spreads very fast. As a result, people’s perception to food safety risks may be an exaggeration of the current food safety situation, causing the decline in social trust and affecting social stability. Therefore, it is of great practical significance to study the influence of Internet use on people’s food safety awareness in the digital era. However, the existing research has not directly studied (to the best of the authors’ knowledge) the relationship between Internet use and people’s food safety perception or evaluation, especially the empirical research in developing countries.

The purpose of this paper is twofold. First, based on the data from China’s national micro-survey (China Social Survey or CSS), the present study provides an overview of the public’s perceptions and evaluations about food safety in China for the years 2013 and 2015. In addition, a comparison of food safety perceptions and evaluations between netizens and non-netizens is made. Second, econometric methods are employed to empirically investigate the association between Internet use and food safety evaluation. Compared with the existing literature, the outstanding contributions of this paper are reflected in the following aspects: 1) This paper has investigated the correlation between technical factors, Internet use, and public food safety perceptions in China and further examined the associations with age, urban–rural, and gender differences, as well as the correlations among Internet use frequencies, attitudes towards the Internet, and people’s food safety evaluation, which expands the scope of research on factors influencing food safety perception. 2) This paper also contributes to studies on the non-economic impact of Internet use on human development. 3) The research conclusions of this study have practical significance, which has important policy implications for food safety risk management in the digital era.

The rest of this article is organized as follows. Section 2 presents the research background, data, and sample introduction. The econometric model is set in Section 3. Section 4 mainly reports the empirical results. Finally, the Section 5 and Section 6 present the relevant discussion and conclusions in this paper respectively.

## 2. Background, Data, and Sample Introduction

### 2.1. Food Safety and Internet Use in China

Like many other developing countries, residents’ consumption level in China has increased dramatically during the past few decades. Mainland China’s per capita consumption expenditure rose from 3206 yuan in 1993 to 52,912 yuan in 2017, with an average annual increase of 12.4% [39]. However, in recent years, the frequent occurrence of various food safety incidents in China (e.g., the case of toxic Sanlu Milk Powder in 2008, the illegal cooking oil incident in 2010, and cadmium contamination in rice in 2013) has greatly increased public concerns about China’s food safety situation, posing overwhelming challenges to improve the quality of food safety in China [40,41]. China is not only a country with the largest food consumer population in the world, but is also a country with a great amount of food exports to other countries. Therefore, China’s food safety also plays a decisive role in the world’s food safety [25].

China’s worrying food safety situation has attracted wide attention from scholars. These studies mainly focus on three aspects. The first branch mainly analyzes the current situation of Chinese consumers’ food safety perceptions [10,41,42,43,44,45]. The second aspect focuses on the determinants of the public perception of food safety, e.g., Liu and Ma [25] and Yang et al. [46]. The third aspect is mainly about the current situation of food safety management and the effective measures to promote food safety management in China, including Yang et al. [47], Wen et al. [48], Xiong et al. [49], Unnevehr and Hoffmann [50], etc.

Interestingly, even though China is a developing country and only began to introduce Internet technology in the 1990s, the speed of Internet development in China is very high. By the end of 2017, the number of Internet users (including mobile Internet) in China had reached 771.98 million, with an Internet penetration of 55.53%, making China the county with the largest Internet usage in the world. In addition, China has already been a global leader in many areas of Internet application, such as mobile payment and big data. Nowadays, the Internet has been closely intertwined with the daily life of Chinese residents, e.g., people can learn about the latest developments in the world through the Internet, express their opinions on social issues, and participate in social interaction. Conversely, the network is also shaping and changing individual values and behaviors while transmitting information to people [51,52,53,54,55,56,57].

Thanks to the convenience of the network in disseminating information and its widespread influence on the audience, it has become an indispensable channel to report food-safety-related information in China. By analyzing a consumer survey in Beijing and Baoding, Liu et al. [41] found that the Internet had become the second-most frequently used channel for consumers to access food-safety-related information. Relevant data showed that in 2016, China’s mainstream Internet public opinion had reported 18,614 food safety incidents, with an average of 51 per day [58]. In addition, the Internet also plays an important role in the food safety management. Liu et al. [59] found that the application of Internet of Things (IoT) technology is conducive to improving food security. There is no doubt that with the further popularity of the Internet in China, its impact on food safety will be more extensive in the future.

### 2.2. Data

The data employed in this study came from the recent China Social Survey (CSS) [60]. The CSS is a well-known national cross-sectional survey project, which has been hosted by the Institute of Sociology, Chinese Academy of Social Sciences, since 2005. The purpose of the CSS is to obtain data about China’s social changes in the transition period through a long-term survey of the national public’s employment, family life, and social attitudes, which has become an important source for scholars to study social issues in China (e.g., Zhang et al. [34]). The main objective of this paper is to analyze the current situation of Chinese people’s food safety perceptions and investigate the association between Internet use and food safety evaluations. At present, the published CSS data only include food safety perception data in 2013 and 2015; hence, this paper finally selected CSS in 2013 and 2015 (i.e., CSS 2013 and CSS 2015). CSS 2013 and CSS 2015 had interviewed 10,206 and 10,243 people, respectively, covering 30 provincial-level administrative units in mainland China (excluding Xinjiang, Hong Kong, Macao, and Taiwan).

### 2.3. Sample Description 

There were two questions in the CSS 2013 and CSS 2015 related to respondents’ attitudes toward food safety. The first question was to ask respondents: “Do you think food safety is a major social problem facing China today?” (both in CSS 2013 and CSS 2015). This paper employed respondents’ subjective response to this question to measure food safety perception, where 1 = yes, 0 = no. The second question was to ask respondents about their subjective evaluation of food safety on a scale of 1–4 (only in CSS 2013), where 1 = very unsafe, 2 = unsafe, 3 = safe, 4 = very safe. We defined respondents’ answers to the second question as a food safety evaluation. Therefore, the higher the value is, the lower the respondents’ risk perceptions about food safety would be.

Table 1 reports the average of the netizens’ and non-netizens’ food safety perception in 2013 and 2015. First, in the 2013 round, the proportion of respondents who believed that China had potential food safety hazards was 18.11%, significantly less than the 22.38% in 2015 (this has been verified using *t*-tests). Moreover, the food safety perception in 2013 for all sub-samples was also significantly less than that in 2015, indicating that food safety worries among Chinese residents had increased from 2013 to 2015. Second, there were significant differences in respondents’ food safety perceptions in the subdivided samples. Specifically, women were more worried about food safety than men. Compared with rural residents, urban residents were more worried about food safety. In addition, food safety perception also presented significant differences among different age groups, namely respondents younger than 31 were more concerned about food safety, and respondents older than 44 were the least worried about food safety. Third, there were obvious differences in food safety perceptions between netizens and non-netizens. Interestingly, in all the sub-samples, netizens were more worried about food safety than non-netizens.

Table 2 shows the average food safety perception at the provincial level in China. In the 2013 round, the four provinces with the highest food safety perceptions were Shanghai, Beijing, Hainan, and Tianjin, with the proportions that considered food safety to be a major social problem were 32.87%, 29.55%, 27.69%, and 25.76%, respectively. In the 2015 round, Guizhou, Tianjin, Shanghai, and Fujian were the four provinces with the highest food safety risk perceptions. In general, the provinces with the highest food safety perceptions were mainly the economically developed eastern provinces in China. In addition, food safety perception increased in 21 of the 30 provinces surveyed, and decreased in 9 provinces from 2013 to 2015.

Table 3 reports the average of the netizens’ and non-netizens’ food safety evaluations in 2013. It can be seen that the average food safety evaluation for the total sample in 2013 was 2.3918, namely between “unsafe” and “safe”. Analogously, there were obvious differences in food safety evaluation among the subdivided samples. In addition, the netizens’ food safety evaluations were lower than that of non-netizens in all sub-samples.

Figure 1 depicts the distribution of netizens’ and non-netizens’ food safety evaluations. It can be seen that compared with non-netizens, netizens reported a higher proportion of “very unsafe” and “unsafe.” As a corollary, the proportions of “safe” and “very safe” reported by netizens were significantly lower than that of non-netizens.

Summarizing the above analysis, the following two conclusions can be drawn. First, at the levels of the full sample, sub-sample, or region, Chinese residents’ concerns about food safety had increased from 2013 to 2015. Second, netizens were more worried about food safety and had higher negative evaluations of food safety. Was the change of food safety perception among Chinese residents and their evaluation on food safety affected by Internet use? To answer this, the rest of this paper attempts to use econometric methods to further analyze the relationship between the Internet use and people’s food safety evaluations.

## 3. Methodology and Variable Selection 

In order to investigate the association between Internet use and people’s food safety evaluations, this paper constructs the following econometric model, shown in the Equation (1). It should be noted that the food safety perception in CSS only reflected whether the respondents thought food safety was the most important social problem in China. However, the Internet use may also have an effect on people’s perception to others social problems. Therefore, the relationship between Internet use and food safety perception has not been further discussed in the empirical section.
(1)$$Y_{i}=\beta_{0}+\beta_{1}Internet\;use_{i}+\gamma X+\delta_{i}Z_{i}+\varepsilon_{i}$$
where *i* represents the *ith* respondent. *Y_i_* is the dependent variable (food safety evaluation). *Internet use_i_* is the key explanatory variable, which is a binary dummy variable (i.e., if the respondent is a netizen, *Internet use_i_* is equal to 1, otherwise it is 0). The *β_1_* is the main estimated parameter, which reflects the association between Internet use and people’s food safety evaluation. The vector *X* represents a series of control variables that may affect food safety evaluation. Referring to Huang and Peng [10] and Ha et al. [19], the control variables mainly include gender, age, education, political identity, household registration, marital status, family economic status, and wellbeing. *Z_i_* is a vector including province dummies. Finally, *ε_i_* is the error term.

It should be noted that we have deleted all missing values for each variable. In addition, samples with respondents’ uncertain answers (e.g., answered “unclear” or “unknown”) were also excluded. As discussed above, we only used CSS 2013 because of the data unavailability. Overall, 9536 samples were obtained. Table 4 reports the definitions and descriptive statistics for the main variables. It can be seen that the proportion of netizen samples was 31.58%, the proportion of the urban sample was 55.36%, and the proportion of male samples was 44.92%. In addition, we have examined the correlation matrix among variables and found that the correlation coefficient between any two variables was no more than 0.6. Finally, we further checked the variance inflation factor (VIF) among variables, and the mean and highest VIF were 1.41 and 1.93, respectively. Therefore, in this paper, the multicollinearity problem was not serious.

In order to preliminarily test the correlation between Internet use and food safety evaluations, we first produced a scatter diagram between the average Internet use and food safety evaluation at the provincial level (see Figure 2). It shows that there was a negative correlation between them, namely that food safety evaluation tended to be lower in provinces with higher Internet use.

## 4. Empirical Results

### 4.1. Benchmark Model Regression Results

Considering that food safety evaluation was a 1–4 ordered variable, we used an ordered probit regression model to estimate the food safety evaluation equation. The estimation results of the benchmark model are shown in Table 5. As a comparison, we also report the results of ordinary least squares (OLS) estimation in columns (1)–(3). The *R*^2^ for columns (1)–(3) increased successively, indicating that the model was set appropriately. Specifically, from the results of the ordered probit regression, the coefficients of Internet use in columns (4)–(6) were all significantly negative (at the 1% level), indicating there was a negative correlation between Internet use and individuals’ food safety evaluation, i.e., compared with non-netizens, netizens had a lower probability of rating food safety as “very safe.” Therefore, the regression results of the benchmark model showed that Internet users were more likely to have a lower food safety evaluation in China, which was consistent with the statistical description results given above.

The relationships between control variables and food safety cognition were different. Specifically, from the results in columns (4)–(6), compared with urban residents and women, rural residents and men had higher evaluations regarding food safety. Education was significantly correlated with the food safety evaluation, which showed that people with a college education and above were more worried about food safety. In addition, people with higher happiness were more satisfied with food safety. However, age, family economic status, and marital status were all found to be not significantly correlated with food safety evaluation in our study.

### 4.2. Further Analysis

#### 4.2.1. Sub-Sample Analysis

Through the above analysis, we found that, on the whole, Internet users were more likely to have a lower food safety evaluation in China. However, results in Table 5 also highlight a puzzle. Specifically, as shown in Table 1, Table 2 and Table 3, there were obvious differences in food safety perceptions and food safety evaluations among different age groups, but the results in Table 5 show that there was no significant correlation between age and food safety evaluation. Therefore, in order to verify the reliability of the above conclusions, we further divided the sample into four categories according to the age of the respondents. 

In addition, the association between Internet use and food safety evaluation may exist through gender and urban–rural differences. On the one hand, influenced by traditional culture, the family work regarding cooking is still mainly undertaken by women in China. Therefore, they may pay more attention to information related to food safety. On the other hand, a large part of the food consumed by rural residents is mainly supplied by themselves. Moreover, cities are also the major outbreak areas for foodborne diseases and food safety scandals. As a result, compared with rural residents, urban residents may be more sensitive to food safety. Therefore, we further subdivided the sample by gender and the urban–rural divide. Table 6 and Table 7 show regression results regarding the associations between Internet use and food safety evaluations for the subdivided samples.

As shown in Table 6 and Table 7, first, the association between Internet use and food safety evaluation was significantly different among people at different ages. Specifically, Internet use had a significant negative correlation with food safety evaluation for people under 65 years old, which was consistent with the results of the benchmark model. However, Internet use had no significant correlation with food safety evaluations for people over 64 years old. Those results are in line with Slavny et al. [61], Jones et al. [62], and Kisley et al. [63], indicating that there are age differences in individuals’ cognitive biases, and older people are more likely to focus on positive information than younger people, also known as the “positivity effect” [64]. Second, in terms of gender, Internet use had a significant negative correlation with food safety evaluation for both men and women. Analogously, in terms of urban and rural, Internet use also had a significant negative correlation with food safety evaluations both for urban and rural residents. However, contrary to our expectations, the negative correlation between Internet use and food safety evaluation for male and rural residents was more obvious. The possible reason is that there are gender differences in people’s attentional bias to different online information. For example, Kinney et al. [65] found that compared with men, women have a larger attentional bias to positive words. Moreover, relative to urban residents, rural residents are less educated and lack the rational identification ability toward network information. As a result, rural residents may be more vulnerable to the negative information on the Internet.

#### 4.2.2. Internet Use Frequency (Content), Attitude towards Internet, and Food Safety Evaluations

Generally speaking, people have certain differences in the frequency and purpose of Internet use, which may ultimately affect the association between Internet use and food safety evaluations. Therefore, according to the three questions in CSS 2013, namely “how often do you use the Internet to browse news?,” “how often do you use the Internet to find information?,” and “how often do you use the Internet to browse Weibo?,” we constructed three variables denoted by “frequency of using Internet to browse news,” “frequency of using Internet to find information,” and “frequency of using Internet to browse Weibo,” respectively. The above three variables were all discrete variables of 1–6, where 1 = never, and 6 = every day.

In addition, if Internet use affects people’s food safety evaluation because of access to information related to food safety on the Internet, will people’s attitude toward the Internet also affect their food safety evaluation? Therefore, according to the three questions in CSS 2013, namely “do you think that the news on the Internet is not as credible as TV, radio, and newspapers?,” “do you think that the Internet is the best channel to express public opinion and reflect the real situation of society?,” and “do you think that netizens are only a small part of the citizens and their opinions cannot represent all citizens?,” we constructed three variables that reflected respondents’ attitudes toward the Internet, which were denoted by “attitudes towards Internet information,” “attitudes towards the Internet reflecting public opinion,” and “attitudes towards netizens,” respectively. Respondents answered the first and third questions on a scale of 1–4, where 1 = strongly disagree, 2 = disagree, 3 = agree, and 4 = strongly agree. For the second question, the respondents answered “strongly agree,” “agree,” “disagree,” “strongly disagree,” which were assigned values of 1–4, respectively. Therefore, there will be more respondents’ suspicions of Internet information or netizens’ opinions with a greater assignment of the above three variables. The regression results of the relationships among Internet use frequencies (content), attitudes toward the Internet, and food safety evaluation are shown in the Table 8. 

The results in Table 8 show that the three indicators reflecting the frequencies of Internet use all had significant negative correlations with food safety evaluations, indicating that the increasing frequencies of using the Internet to browse news, search for information, and browse Weibo would reduce people’s evaluations of food safety. Simultaneously, the coefficients of the three variables reflecting people’s attitudes toward the Internet were all significantly positive in Table 8, indicating that the more doubts people had about Internet information or netizens’ opinions, the higher their food safety evaluation was. 

As discussed above, the negative correlations between Internet use and the food safety evaluation of rural residents, young people, and people who blindly believe in Internet information are more obvious, which may be because those people lack a more objective attitude toward information from the Internet. Therefore, in order to verify the above conclusions, this paper further subdivided the samples according to the individuals’ education level. Generally speaking, people with a higher education level can have a higher comprehension and inclusiveness in the face of different types of information. For example, van Elsas [66], Nie et al. [67], and Golebiowska [68] suggested that people with a higher education level have a better political understanding. Borgonovi [69] also found that better education is associated with higher trust and more tolerance toward migrants. Specifically, we divide the sample into the following two parts according to the education level of the respondents, namely below senior high school and senior high school or above, and the results are shown in the Table 9.

The results in Table 9 indicate that the influence coefficients of Internet use on the residents’ food safety evaluation with different education levels were all significantly negative. However, compared with people with an education level of senior high school or above, Internet use had a stronger negative correlation with food safety evaluations for people with education below a senior high school level. The results further confirmed the previous conclusion, indicating that Internet use had a greater negative impact on people who lacked a rational view of network information.

### 4.3. Robustness Check

#### 4.3.1. Propensity Score Matching Analysis

In this paper, Internet use may cause self-selection problems, e.g., personal Internet use behavior is closely related to their own education level, income, and the area they live in. In addition, because the initial conditions of netizens and non-netizens are different, selection bias may exist in the estimation results. In order to reassure our results were robust and reliable, we further used a propensity score matching (PSM) method to deal with the selection bias. PSM was proposed by Rosenbaum and Rubin [70,71], which mainly solves the selection bias of a sample by constructing a counterfactual framework. Nowadays, PSM has been widely used in the social sciences, including economics, demography, and sociology, such as in Waibel et al. [72], Choi et al. [73], and Cox-Edwards and Rodríguez-Oreggia [74].

Specifically, in this paper, the main steps to carry out PSM were as follows. First, taking Internet use as the dependent variable (i.e., the treatment variable and denoted by D), we employed a probit model to screen matching variables and calculated the propensity score value. Second, according to the propensity score value, samples with the same or similar characteristic conditions were selected from the control group (D = 0) and matched with the treatment group (D = 1) to obtain a hypothetical control group. Finally, the average treatment effect on the treated (ATT) was obtained by comparing the hypothetical control group with the treatment group as follows:(2)$$\begin{array}{ll} ATT & =E[Y_{1}-Y_{0}|D=1]\\ & =E\{E[Y_{1}-Y_{0}|D=1,P(X)]\} \end{array}$$
where *Y*_1_ and *Y*_0_ represent the food safety evaluation of the matched samples in the treatment group and the control group, respectively, and *P*(*X*) is the probability of respondents using the Internet (i.e., propensity score value). In this study, four matching methods were used to obtain the ATT respectively, namely two-nearest neighbor matching, radius matching, kernel matching, and local linear regression matching. As shown in Table 10, ATT was significantly negative in all four matching methods, ranging from −0.1137 to −0.1015, which were also very close to the estimation results under OLS in Table 5.

#### 4.3.2. Controlling City Fixed Effects

Although we have controlled for several variables in the benchmark model that may affect individuals’ food safety evaluations, there may have been some other unobservable city-level (or unreachable) variables that could have affected both food safety evaluation and Internet use, which could cause endogenous problems. For example, public concerns about food safety are often closely related to the size of cities, and big cities usually have a higher Internet penetration. Therefore, this paper further adds a series of city dummy variables (replace the province dummy variables) into the model to control for some unobservable city fixed effects. In addition, we also report robust standard errors clustered by cities in all results, as shown in Table 11. The results show that after controlling for the city fixed effects, the associations between Internet use and food safety evaluations of different populations were still consistent with the benchmark model.

## 5. Discussion

Analyzing the influencing factors on consumers’ food safety perceptions and evaluations has always been a critical branch of food safety related research. Previous studies have shown that personal exposure to food-safety-related information can affect their food safety perception [19]. In the digital era, the Internet has become the most important channel for people to get information. As a result, the significance of the Internet in disseminating food safety knowledge and in managing food safety has received sustained attention. However, evidence on whether Internet use would directly affect individuals’ food safety evaluation is still inadequate, especially in developing countries like China, where food safety incidents have occurred frequently and have simultaneously experienced rapid Internet coverage during the past few years. Based on the Chinese Social Survey for 2013 and 2015, this paper mainly focused on two aspects: 1) to analyze the current situation and change trends of Chinese residents’ food safety perceptions and evaluations, and 2) to empirically investigate the relationship between Internet use and people’s food safety evaluations.

From the statistical analysis, we have found that 18.11% of those surveyed in 2013 thought that food safety had become the main social problem facing China, but this proportion had increased to 22.38% in 2015. At the provincial level, we found that residents in 21 of the 30 provinces surveyed were more worried about food safety in 2015 than 2013. In addition, the subdivided samples by gender, urban–rural divide, and age also showed that the level of Chinese residents’ concerns about food safety had increased from 2013 to 2015. The above results are in line with the finding of previous related studies. For example, Huang and Peng [10] studied the perceptions regarding GM food safety among Chinese urban residents and found that the level of their worries about GM food increased by 30% during 2002 to 2012.

In order to explain the reasons for the increase of food safety concerns among Chinese residents, in the regression analysis section, this study used an ordered probit model to empirically investigate the association between Internet use and people’s food safety evaluations. We also controlled for several personal characteristic factors that may affect food safety evaluations in the benchmark model, including gender, age, education, political identity, household registration, marital status, family economic level, wellbeing, and province dummies. The results show that compared with non-netizens, netizens had a higher probability of rating food safety as “very unsafe.” Therefore, the empirical results indicate that Internet use may be a factor contributing to the increase in food safety concerns among Chinese residents during the past few years.

Negativity bias effect can contribute toward understanding the internal mechanism of Internet use’s impact on people’s food safety evaluations. Compared with positive information, negative news related to food safety would attract more attention from netizens and have a stronger impact on their cognition. Moreover, in order to get more clicks, web editors and Internet media may also prefer the coverage of scandals related to food safety, which may also exacerbate the spread of food safety rumors.

In this paper, two measures have been applied to further verify the reliability of the above conclusions. First, we further investigated relationships among people’s frequencies of different Internet use activities, attitudes toward the Internet, and food safety evaluations. The results show that the more frequently people used the Internet to browse news, find information, and browse Weibo, the lower their evaluations of food safety were. However, those who were skeptical of Internet information and netizens’ opinions had a higher assessment of food safety. Second, we employed the PSM method to deal with the selection bias caused by Internet use, and controlled city fixed effects to further reduce the endogenous problems caused by the omitted variables at the city level. In general, our study still suggests that netizens were more likely to worry about food safety in China.

The associations between Internet use and food safety evaluation had an obvious heterogeneity among different demographic groups. Specifically, Internet use had a stronger negative correlation with food safety evaluation for people under 65 years old, but had no significant correlation with food safety evaluation for people over 64 years old. The possible reason is that people over 64 are less likely to use the Internet, and in general, older people are more rational than younger people when facing problems, thus less affected by negativity bias. We also found that Internet use had a significant negative correlation with food safety evaluation for both men and women. However, contrary to our initial expectation, Internet use had a stronger negative correlation with food safety evaluation for men, which may be caused by differences in the focus of attention on food safety information on the Internet between men and women. There was a negative correlation between Internet use and food safety evaluation for both urban and rural residents. Compared with urban residents, the negative correlation between Internet use and food safety evaluation for rural residents was more obvious. Holtkamp et al. [27] found that food safety reports in China increased with the rate of urbanization. They hold that compared with rural areas, the food system in cities is more complex, which increases the bias of the media in reporting food safety. However, as we found regarding CSS 2013, the overall level of education for rural residents was significantly lower than that of urban residents. As a result, rural residents lacked rational judgment on the negative information related to food safety. Therefore, this paper further divided the sample into two parts in terms of education level: Below senior high school and senior high school or above. The results indicate that Internet use had a stronger negative correlation with food safety evaluation for people with education below a senior high school level.

Consistent with Liu and Ma [25], this paper also found that several demographic variables had important associations with food safety evaluations. Specifically, women, urban residents, and residents with higher education were more worried about food safety, while people with higher happiness were more satisfied with food safety. Nevertheless, in this study, the direct relationships among family economic status, marital status, and food safety evaluations were not proven. Therefore, whether these variables have a moderating effect on the relationship between Internet use and food safety evaluation deserves further study in the future.

This paper further enriches the food safety related research, and contributes to the evidence about the association between the media and people’s food safety perceptions. In recent years, the role of the Internet in food safety management has received sustained attention worldwide. However, countries should also pay attention to the possible impact of the Internet on people’s social cognition. Internet can aggravate the spread of food safety rumors and increase people’s pessimism regarding food safety because of negativity bias, which ultimately affects social sustainability. In particular, those who are less educated and lack rational cognition toward online information are more susceptible to negative information related to food safety. Therefore, on the one hand, in the digital era, the government should deal with various food safety incidents in a timely and effective manner, and do a good job in dispelling rumors about food safety information. Otherwise, the network can amplify the scope and intensity of rumor propagation, and can negatively affect the honest workers in the food industry. On the hand, citizens should be appropriately guided to view Internet information more rationally. For example, it is necessary to continue to invest in education in developing countries and strengthen the popularization of food-safety-related knowledge.

Several limitations of this paper need to be further studied in the future. First, due to the data unavailability, this paper only examined the correlations between Internet use and food safety evaluations, which does not necessarily show a causal link. Although we tried to use PSM to identify the causal relationship between Internet use and food safety evaluations, and controlled a series of covariables to reduce the effect of the omitted variables, endogeneity may exist because of other unobservable individual characteristic variables and reverse causality in this study. We could not draw conclusions about whether lower food safety evaluations were due to Internet use. Therefore, it is necessary to use panel data to further investigate the impact of Internet use on food safety evaluation in the future. In addition, the mechanism of that impact is also worthy of in-depth analysis, e.g., the manifestation on netizens’ negativity bias toward food-safety-related information. Moreover, this paper only discussed the food safety perception in 2013 and 2015, and only employed the data from CSS 2013 in the empirical section. Therefore, future research can utilize the latest data to examine the changing trend of public food safety perceptions in China over a long period of time and provide the latest evidence regarding the impact of Internet use and food safety perceptions. Third, future studies to investigate the differences of food safety perception and the relationship between Internet use and food safety perception across countries also have important practical significance for global food safety risk management. Finally, the dependent variable in this paper was the overall evaluation of food safety by the respondents. Therefore, we encourage future research to examine the impact of Internet use on people’s safety perceptions of specific foods, such as genetically modified (GM) food safety or imported food safety.

## 6. Conclusions

In this study, we found that there is a significant negative correlation between Internet use and people’s food safety evaluation in China. In addition, Internet use has a stronger negative correlation with food safety evaluation for those lacking rational judgment regarding Internet information. These findings were still robust when we consider potential endogeneity and self-selection problems.

## Figures and Tables

**Figure 1 ijerph-16-04162-f001:**
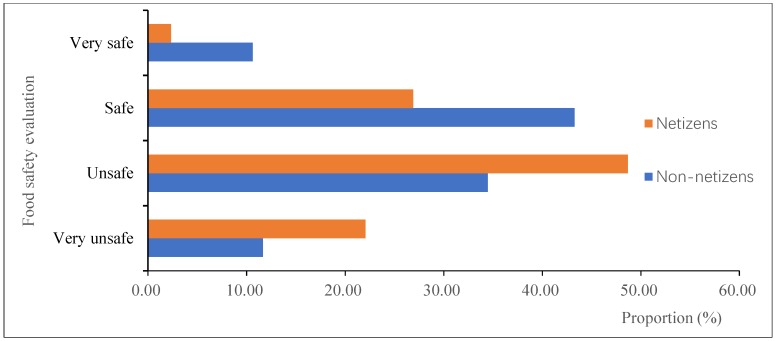
The distribution of food safety evaluations by netizens and non-netizens. Note: Data were from Chinese Social Survey [60] for 2013.

**Figure 2 ijerph-16-04162-f002:**
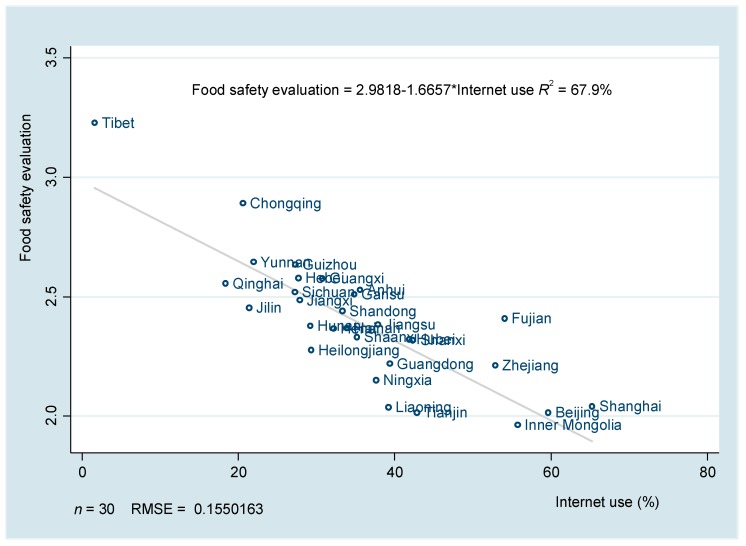
Relationship between Internet use and food safety evaluation at provincial level.

**Table 1 ijerph-16-04162-t001:** Average food safety perceptions for netizens and non-netizens (%).

Year	Sample Category	Overall	Non-Netizens	Netizens
2013 (N = 9536)	Whole	18.11	14.04	26.93
	Female	18.74	14.78	29.05
	Male	17.34	13.01	24.95
	Rural	12.26	10.88	21.07
	Urban	22.83	18.12	28.33
	Age < 31	23.35	15.25	25.58
	30 < Age < 45	20.48	15.28	28.27
	44 < Age < 65	15.08	13.14	27.67
	Age > 64	15.30	15.10	20.69
2015 (N = 10048)	Whole	22.38	16.41	31.93
	Female	23.12	17.18	33.92
	Male	21.49	15.38	29.92
	Rural	15.79	13.44	24.04
	Urban	27.89	20.43	34.74
	Age < 31	28.16	19.54	29.19
	30 < Age < 45	27.51	18.19	34.87
	44 < Age < 65	18.76	15.73	31.96
	Age > 64	17.14	16.49	26.56

Note: Data were from Chinese Social Survey [60] for 2013 and 2015.

**Table 2 ijerph-16-04162-t002:** Average food safety perceptions among provinces in China (%).

Province	Overall	2013	2015	Change	Province	Overall	2013	2015	Change
Anhui	24.24	20.81	27.59	+	Jiangxi	19.53	14.62	23.89	+
Beijing	27.31	29.55	25.00	−	Liaoning	21.94	23.53	20.35	−
Fujian	24.52	18.73	29.96	+	Jiangsu	19.04	15.86	22.07	+
Gansu	16.05	19.12	12.94	−	Ningxia	21.05	19.70	22.39	+
Guangdong	24.05	21.62	26.20	+	Qinghai	15.27	9.52	20.59	+
Guangxi	14.68	14.91	14.46	−	Shandong	20.03	19.05	20.91	+
Guizhou	20.73	7.84	33.08	+	Shanxi	21.17	19.41	22.91	+
Hainan	24.62	27.69	21.54	−	Shaanxi	21.91	19.88	23.89	+
Hebei	9.83	11.94	7.82	−	Shanghai	31.91	32.87	30.94	−
Henan	18.39	17.28	19.46	+	Sichuan	16.34	13.73	18.80	+
Heilongjiang	14.18	14.02	14.34	+	Tianjin	28.41	25.76	31.06	+
Hubei	17.64	16.49	18.75	+	Tibet	17.19	3.51	28.17	+
Hunan	31.12	25.55	36.48	+	Yunnan	11.11	8.02	13.96	+
Jilin	12.47	14.81	10.29	−	Zhejiang	32.82	26.96	38.40	+
Inner Mongolia	30.38	23.32	37.13	+	Chongqing	8.42	9.73	7.18	−
Total	20.30	18.11	22.38	+					

Note: Data were from Chinese Social Survey [60] for 2013 and 2015. + represents an increase, − represents a decrease.

**Table 3 ijerph-16-04162-t003:** Average food safety evaluation for netizens and non-netizens.

Year	Sample Category	Overall	Non-Netizens	Netizens
2013	Whole	2.3918	2.5284	2.0956
	Female	2.3978	2.5292	2.0549
	Male	2.3845	2.5273	2.1338
	Rural	2.6704	2.7221	2.3420
	Urban	2.1671	2.2782	2.0370
	Age < 31	2.2244	2.5169	2.1439
	30 < Age < 45	2.2822	2.4411	2.0444
	44 < Age < 65	2.4915	2.5541	2.0841
	Age > 64	2.5846	2.6000	2.1724

Note: Data were from Chinese Social Survey [60] for 2013.

**Table 4 ijerph-16-04162-t004:** Definitions and descriptive statistics of the main variables.

Variable	Definition	Mean	SD
Food safety evaluation	How do you evaluate food safety in the current society? Four categories: Very unsafe = 1, unsafe = 2, safe = 3, very safe = 4.	2.3918	0.8349
Internet use	Use the Internet = 1, otherwise = 0.	0.3158	0.4648
Gender	Man = 1, woman = 0.	0.4492	0.4974
Age in 2013			
Age < 31	Yes = 1, else = 0	0.1720	0.3774
45 > Age > 30	Yes = 1, else = 0	0.2944	0.4558
65 > Age > 44	Yes = 1, else = 0	0.4493	0.4975
Age > 64	Yes = 1, else = 0	0.0843	0.2779
Education			
Illiteracy	Illiteracy = 1, else = 0	0.1167	0.3211
Primary school	Primary school = 1, else = 0	0.2514	0.4338
Junior high school	Junior high school = 1, else = 0	0.3243	0.4682
Senior high school	Senior high school (include technical secondary school or vocational high school) = 1, else = 0	0.1721	0.3775
College	College = 1, else = 0	0.1285	0.3346
Graduate	Graduate = 1, else = 0	0.0070	0.0835
Political identity	Whether the respondent is a member of Communist Party of China? Yes= 1, else = 0	0.0998	0.2998
Household registration	Urban = 1, rural = 0.	0.5536	0.4971
Marital status			
Divorced or widowed	Divorced or widowed = 1, else = 0	0.0624	0.2419
In a marriage	In a marriage = 1, else = 0	0.8380	0.3685
Unmarried	Unmarried = 1, else = 0	0.0996	0.2995
Family economic status	What are the local levels of economic and social status of the respondents? Five categories: Low = 1, below medium = 2, medium = 3, above medium = 4, high = 5.	2.3441	0.9054
Well-being	Do you think you are a happy person? Six categories: Strongly disagree= 1 to strongly agree= 6.	4.0841	1.1273

Note: Data were from Chinese Social Survey [60] for 2013. SD—standard deviation.

**Table 5 ijerph-16-04162-t005:** Regression analysis of the relationship between Internet use and food safety evaluation.

Variables	Dependent Variable: Food Safety Evaluation					
	OLS			Ordered Probit		
	(1)	(2)	(3)	(4)	(5)	(6)
Internet use	−0.3815 ***	−0.1085 ***	−0.1173 ***	−0.5159 ***	-0.1513 ***	−0.1648 ***
	(0.0176)	(0.0234)	(0.0234)	(0.0242)	(0.0330)	(0.0331)
Gender		0.0545 ***	0.0616 ***		0.0751 ***	0.0856 ***
		(0.0168)	(0.0167)		(0.0238)	(0.0238)
Age (ref: Younger than 31 years old)						
30 < Age < 45		−0.0447	-0.0380		-0.0664	−0.0572
		(0.0291)	(0.0290)		(0.0412)	(0.0413)
44 < Age < 65		0.0143	0.0135		0.0192	0.0179
		(0.0305)	(0.0305)		(0.0433)	(0.0434)
64 < Age		0.0397	0.0257		0.0544	0.0347
		(0.0402)	(0.0401)		(0.0572)	(0.0574)
Education (ref: Illiteracy)						
Primary school		−0.1598 ***	−0.1650 ***		−0.2283 ***	−0.2370 ***
		(0.0284)	(0.0283)		(0.0410)	(0.0411)
Junior high school		−0.2930 ***	−0.3041 ***		−0.4165 ***	−0.4344 ***
		(0.0292)	(0.0292)		(0.0419)	(0.0421)
Senior high school		−0.4217 ***	−0.4375 ***		−0.5983 ***	−0.6238 ***
		(0.0338)	(0.0339)		(0.0484)	(0.0488)
College		−0.4928 ***	−0.5061 ***		−0.6996 ***	−0.7221 ***
		(0.0392)	(0.0394)		(0.0562)	(0.0566)
Graduate		−0.5919 ***	−0.6118 ***		−0.8539 ***	−0.8858 ***
		(0.1007)	(0.1002)		(0.1501)	(0.1499)
Political identity		−0.0013	−0.0204		-0.0012	−0.0283
		(0.0274)	(0.0275)		(0.0388)	(0.0392)
Household registration		−0.3127 ***	−0.3090 ***		−0.4396 ***	−0.4362 ***
		(0.0186)	(0.0186)		(0.0264)	(0.0265)
Marital status (ref: Divorce or widowed)						
In marriage		0.0070	−0.0206		0.0104	−0.0290
		(0.0333)	(0.0333)		(0.0477)	(0.0479)
Unmarried		0.0685	0.0549		0.0974	0.0780
		(0.0465)	(0.0463)		(0.0663)	(0.0664)
Family economic status			0.0148			0.0209
			(0.0093)			(0.0132)
Well-being			0.0573 ***			0.0826 ***
			(0.0079)			(0.0113)
Province fixed effect	YES	YES	YES	YES	YES	YES
*R* ^2^	0.0970	0.1607	0.1672			
N	9536	9536	9536	9536	9536	9536

Note: *, **, and *** represent 10%, 5%, and 1% levels of statistical significance, respectively. Robust standard errors are reported in parentheses. OLS: Ordinary least squares. Data were from Chinese Social Survey [60] for 2013.

**Table 6 ijerph-16-04162-t006:** Regression analysis of the relationship between Internet use and food safety evaluation: Sub-sample analysis.

Variables	Dependent Variable: Food Safety Evaluation (Ordered Probit)			
	Age < 31	30 < Age < 45	44 < Age < 65	Age > 64
	(1)	(2)	(3)	(4)
Internet use	−0.1872 **	−0.2100 ***	−0.1927 ***	−0.0255
	(0.0817)	(0.0534)	(0.0552)	(0.2000)
Control variables	YES	YES	YES	YES
Province fixed effect	YES	YES	YES	YES
N	1640	2807	4285	804

Note: *, **, and ***represent 10%, 5%, and 1% levels of statistical significance respectively. Robust standard errors are reported in parentheses. Data were from Chinese Social Survey [60] for 2013.

**Table 7 ijerph-16-04162-t007:** Regression analysis of the relationship between Internet use and food safety evaluation: Sub-sample analysis.

Variables	Dependent Variable: Food Safety Evaluation (Ordered Probit)			
	Male	Female	Urban	Rural
	(1)	(2)	(3)	(4)
Internet use	−0.1938 ***	−0.1400 ***	−0.1501 ***	−0.2397 ***
	(0.0478)	(0.0465)	(0.0403)	(0.0612)
Control variables	YES	YES	YES	YES
Province fixed effect	YES	YES	YES	YES
N	4284	5252	5279	4257

Note: *, **, and *** represent 10%, 5%, and 1% levels of statistical significance, respectively. Robust standard errors are reported in parentheses. Data were from Chinese Social Survey [60] for 2013.

**Table 8 ijerph-16-04162-t008:** Internet use frequencies (content), attitudes toward Internet, and food safety evaluations.

Variables	Dependent Variable: Food Safety Evaluation (Ordered Probit)					
	(1)	(2)	(3)	(4)	(5)	(6)
Frequency of using Internet to browse news	−0.0307 **					
	(0.0144)					
Frequency of using Internet to find information		−0.0580 ***				
		(0.0130)				
Frequency of using Internet to browse Weibo			−0.0367 ***			
			(0.0117)			
Attitudes toward Internet information				0.0527 *		
				(0.0303)		
Attitudes toward the Internet reflecting public opinion					0.0615 **	
					(0.0299)	
Attitudes toward netizens						0.0556 *
						(0.0295)
Control variables	YES	YES	YES	YES	YES	YES
Province fixed effect	YES	YES	YES	YES	YES	YES
N	3002	3002	2995	2864	2878	2921

Note: *, **, and *** represent 10%, 5%, and 1% levels of statistical significance, respectively. Robust standard errors are reported in parentheses. Data were from Chinese Social Survey [60] for 2013.

**Table 9 ijerph-16-04162-t009:** Regression analysis of the relationship between Internet use and food safety evaluations: Different education levels.

Variables	Dependent Variable: Food Safety Evaluation (Ordered Probit)	
	Below Senior High School	Senior High School or Above
	(1)	(2)
Internet use	−0.2333 ***	−0.2123 ***
	(0.0424)	(0.0546)
Control variables	YES	YES
Province fixed effect	YES	YES
N	6603	1641

Note: *, **, and *** represent 10%, 5%, and 1% levels of statistical significance, respectively. Robust standard errors are reported in parentheses. Data were from Chinese Social Survey (CSS) [60] for 2013.

**Table 10 ijerph-16-04162-t010:** Internet use and food safety evaluations: PSM analysis.

Matching Methods	Two-Nearest Neighbor Matching	Radius Matching	Kernel Matching	Local Linear Regression Matching
Average treatment effect on the treated (ATT)	−0.1031 *	−0.1015 *	−0.1137 **	−0.1022 *
	(0.0529)	(0.0524)	(0.0463)	(0.0579)
Control variables	YES	YES	YES	YES
Province fixed effect	YES	YES	YES	YES
Sample number of treatment group	3011	3011	3011	3011
Sample number of control group	6468	6468	6468	6468

Notes: Standard errors are in parentheses. *, **, and *** represent 10%, 5%, and 1% levels of statistical significance, respectively. Data were from Chinese Social Survey [60] for 2013.

**Table 11 ijerph-16-04162-t011:** Regression analysis of the relationship between Internet use and food safety evaluation: Controlling city fixed effects.

Variables	Dependent Variable: Food Safety Evaluation (Ordered Probit)								
	Total Sample	Age < 31	30 < Age < 45	44 < Age < 65	Age > 64	Male	Female	Urban	Rural
	(1)	(2)	(3)	(4)	(5)	(6)	(7)	(8)	(9)
Internet use	−0.1584 ***	−0.1950 **	−0.2196 ***	−0.1623 ***	−0.1106	−0.1815 ***	−0.1340 ***	−0.1533 ***	−0.2267 ***
	(0.0330)	(0.0865)	(0.0572)	(0.0515)	(0.2986)	(0.0512)	(0.0462)	(0.0424)	(0.0701)
Control variables	YES	YES	YES	YES	YES	YES	YES	YES	YES
City fixed effect	YES	YES	YES	YES	YES	YES	YES	YES	YES
N	9536	1640	2807	4285	804	4284	5252	5279	4257

Notes: Robust standard errors clustered by cities are in parentheses. *, **, and *** represent 10%, 5%, and 1% levels of statistical significance, respectively. Data were from Chinese Social Survey [60] for 2013.

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
