# Peer review of "Association of Internet Use with Attitudes Toward Food Safety in China: A Cross-Sectional Study"

_ijerph, 2019, doi:10.3390/ijerph16214162_

Round 1

Reviewer 1 Report

This is an interesting and relevant paper. I like the fact that it addresses perceptions in China. I think the work should be published. It would be great to have a follow up study comparing attitudes in N. America or the EU, regarding food safety in general or even Chinese imports. 

Author Response

Dear reviewers,

  We would like to thank you for your valuable comments. In response to the your comments and recommendations, the paper has been duly revised and the relevant amendments are summarized in the following. In fact, apart from what the reviewers have pointed out, we have also carefully checked this paper and made a few minor changes (especially to some small errors or irregularities) to ensure it to be better. All of the revisions are clearly highlighted in the new revised paper (We mark the changes in yellow).

Reviewer 2 Report

The article is very clear and well written. Some small suggestions:
- Table 1 shows how many participants are in CSS2013 and CSS2015 (N = ...)
- The comment on the data is very clear, but I suggest the authors
to insert in the limits that the data are dated (2013) and to
problematize this aspect. The use of the internet has increased a
lot in recent years so a lot has changed from 2013 to today. A new data
collection would therefore be desirable.

Author Response

Dear reviewers,

  We would like to thank the you for your valuable comments. In response to the your comments and recommendations, the paper has been duly revised and the relevant amendments are summarized in the following. In fact, apart from what the reviewers have pointed out, we have also carefully checked this paper and made a few minor changes (especially to some small errors or irregularities) to ensure it to be better. All of the revisions are clearly highlighted in the new revised paper (We mark the changes in yellow).

Reviewer 3 Report

Title:

Shaping Attitudes Towards Food Safety: The Influence of Internet Use

The paper investigated the current situation of food safety perception and evaluation among Chinese residents and the impact of the Internet use on their food safety evaluation.  I have some concerns and comments.  The authors may find the following comments helpful.

Major Comments:

The causal relationship in the OLS and Ordered Probit (the main analyses) is not clear. Internet use can affect consumers’ knowledge and attitude toward food safety issue. However, authors need to check whether residents become more concerned about food safety as the size of their city is getting bigger due to complexity and increased possibility of contamination on the food distribution process and whether the proportion of netizen is larger as the size of city is bigger due to different accessibility to Internet. If they are true, then both netizen vs. non-netizen and evaluation of food safety can be simultaneously affected by the status of urban vs. rural or the size of city. This paper does not discuss about this.

There are numerous lapses in English usage. The authors need to carefully rewrite the paper in a more concise and clear manner. The first difficulty in this analysis lies in identifying what are the main contributions of the paper, due to unclear presentation and writing.

Some parts are ambiguous. For example, in the Methodology, Variable Selection, the authors need to explain more about the variable of “political identity” and “family economic status and well-being.” Another example is ATT. They need to explain more about ATT. It is too technical.

Analyses in Table 5 are hard to understand. The dependent variable is 4-points Likert scale.  Authors claim that they used OLS and Ordered Probit. They estimated their model differently from the conventional approach. Therefore, they need to explain more about the specification of their estimation model. That is, need to explain what estimation approach they used and why. Results in Table 6 and Table 7 reported the marginal impact only at “very safe.” Need to justify.

Page 13, the second full paragraph. In general, “two years” time points do not say much about a trend. We need at least three time points to discuss a trend from them.

Author Response

Dear reviewers,

We would like to thank you for your valuable comments. In response to the your comments and recommendations, the paper has been duly revised and the relevant amendments are summarized in the following. In fact, apart from what the you have pointed out, we have also carefully checked this paper and made a few minor changes (especially to some small errors or irregularities) to ensure it to be better. All of the revisions are clearly highlighted in the new revised paper (We mark the changes in yellow).

Round 2

Reviewer 3 Report

Title:

 Shaping Attitudes Towards Food Safety: The Influence of Internet Use

Authors made a good revision.

However, a major problem (endogeneity) still remains and therefore the implications are weak. Need to consider changes in writing implications.

Author Response

The authors would like to thank the reviewers for their valuable comments. In response to the reviewers’ comments and recommendations, the paper has been duly revised and the relevant amendments are summarized in the following. In fact, apart from what the reviewers have pointed out, we have also carefully checked this paper and made a few minor changes (especially to some small errors or irregularities) to ensure it to be better. All of the revisions are clearly highlighted in the new revised paper (We mark the changes in red).

Comments and Suggestions from Reviewer 1

A1. Authors made a good revision.

However, a major problem (endogeneity) still remains and therefore the implications are weak. Need to consider changes in writing implications.

Authors’ Responses to Reviewer 1

Dear reviewers,

Thank you for your comment and thank you for your appreciation of the work we have done is this manuscript.

Dear reviewers, indeed, you have pointed out an important weakness in our paper. Due to the data unavailability (this article only employs data from CSS in 2013 to carry out the empirical analysis. However, to the best of our knowledge, among the existing micro surveys in China, only CSS has both respondents' information about food safety perception and Internet use. Moreover, only CSS2013 has respondents' information about the food safety evaluation. Therefore, we only use the CSS2013 in our study), we did not find the appropriate instrumental variables to fully deal with the endogenous problems of the model. However, we still try to reduce the endogenous problem as much as possible by adding the city dummy variables and adopting PSM method.

Yes, as you suggested, although we tried to use PSM to identify the causal relationship between Internet use and food safety evaluation, and controlled a series of covariables to reduce the effect of the omitted variables, endogeneity may exist because of other unobservable individual characteristic variables and reverse causality in this study. Therefore, this paper only examines the correlations between Internet use and food safety evaluation, not necessarily causal. We could not draw conclusions about whether lower food safety evaluation is due to Internet use.

Therefore, we have revised the implications of the paper strictly according to your suggestion. For example, in this paper, we modify the purpose of this study as follows:

The purpose of this paper is twofold. First, based on the data from China’s national micro-survey (China Social Survey or CSS), the present study provides an overview of public perception and evaluation about food safety in China for year 2013 and 2015. In addition, a comparison of food safety perception and evaluation between netizens and non-netizens is made. Second, the econometric methods are employed to empirically investigate the association between Internet use and food safety evaluation.

We modify the title of this study as follows:

Association of Internet Use with Attitudes Towards Food Safety in China: a Cross-sectional Study

We modify the conclusion (abstract) of this study as follows:

In this study, by employing the Chinese Social Survey for 2013 and 2015, we have investigated the current situation of food safety perception and evaluation among Chinese residents and the association between Internet use and individuals’ food safety evaluation. Empirical results indicate that there is a significant negative correlation between Internet use and people’s food safety evaluation in China. Further heterogeneity analysis shows that Internet use has a stronger negative correlation with food safety evaluation for those lack of rational judgment on Internet information. Specifically, the negative correlation between Internet use and food safety evaluation is more obvious among rural residents, young people and less educated residents. Finally, the propensity score matching (PSM) is applied to conduct a robustness check.

We modify the limitations in last part of this study as follows:

Several limitations of this paper need to be further studied in the future. First, due to the data unavailability, this paper only examines the correlations between Internet use and food safety evaluation, not necessarily causal. Although we tried to use PSM to identify the causal relationship between Internet use and food safety evaluation, and controlled a series of covariables to reduce the effect of the omitted variables, endogeneity may exist because of other unobservable individual characteristic variables and reverse causality in this study. We could not draw conclusions about whether lower food safety evaluation is due to Internet use. Therefore, it is necessary to use panel data to further investigate the impact of Internet use on food safety evaluation in the future. In addition, the mechanism of that impact is also worthy of in-depth analysis, e.g., the manifestation on netizens’ negativity bias towards food safety related information. Moreover, this paper only discusses the food safety perception in 2013 and 2015, and only employs the data from CSS2013 in the empirical section. Therefore, future research can update the latest data to examine the changing trend of public food safety perception in China over a long period of time, and provide the latest evidence regarding the impact of Internet use and food safety perception. Third, future studies to investigate the differences of food safety perception and the relationship between Internet use and food safety perception across countries also have important practical significance for global food safety risk management. Finally, the dependent variable in this paper is the overall evaluation of food safety by the respondents. Therefore, we encourage future research to examine the impact of Internet use on people’s perception to specific food safety, such as genetically modified (GM) food safety or importing food safety.

Finally, we have also modified all relevant statements in the text, marked in red.

Thanks again for your valuable advice.
